# C-Reactive Protein Level and the Genetic Variant rs1130864 in the CRP Gene as Prognostic Factors for 10-Year Cardiovascular Outcome

**DOI:** 10.3390/cells12131775

**Published:** 2023-07-04

**Authors:** Susanne Schulz, Selina Rehm, Axel Schlitt, Madlen Lierath, Henriette Lüdike, Britt Hofmann, Kerstin Bitter, Stefan Reichert

**Affiliations:** 1Department of Operative Dentistry and Periodontology, Martin-Luther-University Halle-Wittenberg, 06108 Halle, Germany; selinarehm@aol.com (S.R.); m.lierath@gmx.de (M.L.); h.luedike@gmx.de (H.L.); kerstin.bitter@uk-halle.de (K.B.); stefan.reichert@uk-halle.de (S.R.); 2Department of Cardiology, Paracelsus-Harz-Clinic Bad Suderode, 06485 Quedlinburg, Germany; axel.schlitt@pkd.de; 3Department of Medicine III, Martin-Luther-University Halle-Wittenberg, 06108 Halle, Germany; 4Department of Cardiothoracic Surgery, Martin-Luther-University Halle-Wittenberg, 06108 Halle, Germany; britt.hofmann@uk-halle.de

**Keywords:** cardiovascular disease, C-reactive protein, genetic variants, prognostic factor

## Abstract

Background: Cardiovascular disease (CVD) is the primary cause of premature death and disability worldwide. There is extensive evidence that inflammation represents an important pathogenetic mechanism in the development and prognosis of CVD. C-reactive protein (CRP) is a potential marker of vascular inflammation and plays a direct role in CVD by promoting vascular inflammation. The objective of this study (ClinTrials.gov identifier: NCT01045070) was to assess the prognostic impact of CRP protein levels and genetic variants of CRP gene events on cardiovascular (CV) outcome (10-year follow-up) in patients suffering from CVD. Methods: CVD patients were prospectively included in this study (*n* = 1002) and followed up (10 years) regarding combined CV endpoint (CV death, death from stroke, myocardial infarction (MI), and stroke/transient ischemic attack (TIA)). CRP protein level (particle-enhanced immunological turbidity test) and genetic variants (rs1130864, rs1417938, rs1800947, rs3093077; polymerase chain reaction-restriction fragment length polymorphism (PCR-RFLP) after DNA extraction from EDTA-blood) were evaluated. Results: In survival analyses, increased CRP protein levels of ≥5 mg/L (log-rank test: *p* < 0.001, Cox regression: *p* = 0.002, hazard ratio = 1.49) and CT + TT genotype of rs1130864 (log-rank test: *p* = 0.041; Cox regression: *p* = 0.103, hazard ratio = 1.21) were associated with a weaker CV prognosis considering combined CV endpoint. Conclusions: Elevated CRP level and genetic variant (rs1130864) were proven to provide prognostic value for adverse outcome in CVD patients within the 10-year follow-up period.

## 1. Introduction

Clinical and epidemiological studies are providing increasing evidence that inflammation represents a key factor in the development and prognosis of cardiovascular diseases (CVD) [1]. Several inflammatory mediators have been implicated in this process. C-reactive protein (CRP), a marker of acute phase response, is a very crucial parameter of inflammation involved in noncommunicable diseases, including CVD [2]. CRP level constitutes a predictor for both primary and secondary prevention of atherosclerotic events [3]. Based on this scientific evidence, CRP was included in the guidelines for targeting measures in primary prevention [4].

In a multitude of clinical studies, proof has accumulated that high levels of CRP predict a variety of cardiovascular (CV) outcomes, including CV mortality [5,6,7]. Considering these results, an intervention that selectively inhibits inflammation could improve outcome for CVD patients [8]. Indeed, the clinical efficacy of anti-inflammatory interventions in reducing major adverse cardiac events was demonstrated in different studies [8]. Statins lower the plasma concentrations of atherogenic lipoproteins and CRP and improve CV prognosis [9,10]. Furthermore, direct anti-inflammatory therapies were demonstrated to reduce myocardial infarction (MI), stroke, and CV death [11,12,13,14,15]. 

In addition to an inflammation-related increase in CRP levels, genetic background also accounts for CRP expression [16]. In twin studies, genetic characteristics were proven to play a determining role for about 40% of CRP expression [17,18]. According to genetic association studies 58 loci account for ~7% of the variation in CRP levels (*n* = 200,000 Europeans) [19], and 42 gene sets account for ~16.3% of the variation in CRP levels (*n* = 427,367 and *n* = 575,531 European descent) [20]. Among these genetic markers are the tagging single-nucleotide polymorphisms (SNPs) rs1130864 (3′-UTR, C/T), rs1417938 (intron, T/A), rs1800947 (L184L, G/C), and rs3093077 (noncoding, C/T) [21,22,23]. 

SNPs associated with elevated CRP levels consequently may be accompanied by an increase in CVD risk [24]. However, no causal relationship between genetic background of CRP and the incidence of CVD could be proven by applying mendelian randomization analysis [20,22,25]. In contrast, selected studies have shown a genetic association with the occurrence of CVD [21,26,27]. However, the data are controversial and it is assumed that influencing factors, e.g., ethnicity, affected the study results [28]. 

Furthermore, it would be clinically important to determine whether stable genetic markers of the CRP gene could be considered as predictors of CV outcome. Here, too, the clinical studies yielded controversial results [29,30,31,32,33]. 

In a 3-year follow-up, we demonstrated that GG genotype of rs1800947 and CRP level were associated with adverse CV outcome in a cohort of patients with CVD [21]. The present study (subanalysis of the study: “Periodontitis and Its Microbiological Agents as Prognostic Factors in Patients with Coronary Heart Disease”; ClinicalTrials.gov identifier: NCT01045070) was designed to determine whether the results could be confirmed after a longer follow-up period of 10 years. The evaluation of the study included the potential impact of SNPs of the CRP gene (rs1130864, rs1417938, rs1800947, rs3093077) as well as CRP plasma level on a combined CV endpoint (CV death, death from stroke, myocardial infarction (MI), and stroke/transient ischemic attack (TIA)). 

## 2. Materials and Methods

### 2.1. Study Design

This study represents a subanalysis of the study entitled “Periodontitis and Its Microbiological Agents as Prognostic Factors in Patients with Coronary Heart Disease” (ClinicalTrials.gov identifier: NCT01045070). The investigation was approved by the ethics committee of the Martin Luther University Halle-Wittenberg (18 July 2007). All patients gave their written informed consent. The examinations were carried out in accordance with the ethical guidelines of the “Declaration of Helsinki” and its amendment in “Tokyo and Venice”. This study was conducted as a longitudinal cohort study with a 10-year follow-up period.

### 2.2. CV Patients

#### 2.2.1. Baseline Investigations

Patient characteristics and investigation were described in detail in Reichert et al., 2016 [34]. Briefly: At baseline, 1002 consecutive German patients of Caucasian origin admitted to the Department of Internal Medicine III or Department of Cardiothoracic Surgery of the Martin Luther University Halle-Wittenberg with angiographically proven CVD were prospectively included from October 2009 to February 2011.

Inclusion criteria were the following: age >18 years, number of teeth >4, at least 60% stenosis of one of the main coronary arteries demonstrated by coronary angiography or previous or current percutaneous coronary intervention (PCI) or previous or current coronary artery bypass surgery (CABG). The exclusion criteria were: pregnancy, subgingival scaling and root planning during the last 6 months, antibiotic therapy within the last 3 months, inability to give written informed consent, current alcohol or drug abuse, and intake of drugs that potentially cause gingival hyperplasia.

On the day after inpatient admission and confirmation of inclusion and exclusion criteria, a fasting blood draw was performed in the morning between 7 a.m. and 8 a.m. Biochemical parameters were assessed in the Central Laboratory of the Medical Faculty of the Martin Luther University Halle Wittenberg (CRP: latex particle-enhanced immunological turbidity test; leukocytes: flow cytometry; interleukin 6: electrochemiluminescence immunoassay; uric acid: enzymatic colorimetric assay using uricase; creatinine: kinetic color assay, Jaffé method; total cholesterol: enzymatic colorimetric assay; HDL cholesterol: homogeneous enzymatic colorimetric assay; LDL cholesterol: homogeneous enzymatic colorimetric assay; and triglycerides: enzymatic colorimetric assay). 

On the following day, the study-specific medical history was taken. Baseline variables (age, gender, smoking habits, and body mass index) and patient’s medical history (e.g., diabetes mellitus, hypertension, peripheral arterial disease, dyslipoproteinemia, and medication) were assessed.

#### 2.2.2. Ten-Year Follow-Up Period

Between November 2020 and January 2023, 792 CV patients were included in the 10-year follow-up (dropout rate 21%). The incidence of the predefined combined CV endpoint, including CV death, death from stroke, MI, and stroke/TIA, was calculated. A standardized questionnaire was sent by post for acquiring follow-up data. If patients did not reply, a telephone interview was conducted with the patient or the patient’s relatives and physician (in case of death of the patient). Furthermore, civil registration offices were contacted and information about current address or death date were requested, when it was required.

### 2.3. Genetic Investigations

SNPs in the CRP gene (rs1130864, rs1417938, rs1800947, and rs3093077) were determined. The sequence data were in accordance with the sequence of the human CRP gene (Genbank: accession number NG_013007.1, https://www.ncbi.nlm.nih.gov/search/all/?term=NG_013007; accessed on 28 March 2023). Genotypes were determined by polymerase chain reaction-restriction fragment length polymorphism (PCR-RFLP) after extracting DNA from EDTA blood (QIAamp^®^ blood extraction kit; Qiagen, Hilden, Germany). The amplified products were digested with restriction enzymes (New England Biolabs, according to the manufacturer’s protocols) and size-fractionated by electrophoretic separation in 3% agarose gels. The primers and enzymes used are displayed in Table 1. 

### 2.4. Statistical Procedures

Statistical analyses were carried out using a commercial software SPSS v.25.0 package (IBM, Chicago, IL, USA). Values of *p* < 0.05 were considered as significant. Continuous data (metric demographic, clinical, and serological data) were assessed for normal distribution using the Kolmogorov–Smirnov test and the Shapiro–Wilk test. These data were reported as the median, 25th/75th interquartiles (non-normally distributed values). For statistical evaluation, the Mann–Whitney U test was used. Categorical variables were documented as percentage, applying the chi-squared test for statistical analyses. If the expected values in one group was <5, Fisher’s exact test was performed.

Univariate survival analyses were carried out using Kaplan–Meier curves and the log-rank test. Cox regression was applied in order to generate adjusted hazard ratios (HR).

## 3. Results

### 3.1. Clinical Characterisation of CV Patients

In all, 1002 CV patients were prospectively included in the longitudinal cohort study at baseline, and 792 patients completed the 10-year follow-up (dropout rate: 21%). During the mean follow-up of 378.3 ± 214.8 weeks, 179 CV deaths, 12 deaths from stroke, 89 MIs, and 59 strokes/TIAs were recorded. The total incidence of the combined endpoint was 42.8% (Figure 1).

Baseline characteristics as well as genotype distribution of the CV patients according to the combined endpoint (CV death, death from stroke, MI, and stroke/TIA) are displayed in Table 2. Significant differences were found for age (*p* < 0.001), history of MI (*p* = 0.001), stroke/TIA (*p* = 0.016), peripheral artery disease (*p* = 0.002), and diabetes mellitus (*p* < 0.001) when comparing patients considering CV endpoint during the 10-year follow-up period (Table 2). With respect to biochemical parameters, CV patients meeting the combined endpoint had significantly higher levels of CRP (*p* = 0.001), interleukin 6 (*p* = 0.017), uric acid (*p* < 0.001), creatinine (*p* < 0.001), and total cholesterol (*p* = 0.030), respectively, than patients who did not meet the endpoint. HDL cholesterol was decreased in patients with adverse 10-year outcome (*p* = 0.002).

### 3.2. Circulating CRP Level as Prognostic Markers for Adverse CV Outcome

In this study, in addition to the analysis of genetic markers in the CRP gene, the prognostic significance of CRP levels was also assessed in the 10-year follow-up. As demonstrated by the Kaplan–Meier survival curve and log-rank test, an elevated CRP level is associated with an adverse outcome at 10-year follow-up (*p* _log rank_ < 0.001, HR: 1.54) (Figure 2).

In a multivariate analysis to obtain adjusted HRs, CRP level was proven to be an independent predictor of future CV events. Established CV risk factors such as age, male gender, BMI, smoking, diabetes mellitus, hypertension, and hyperlipoproteinemia were included in this analysis. An increased CRP level could be confirmed as an independent prognostic factor for further CV events (HR: 1.49, 95% CI: 1.16–1.91). In addition, older age, male gender, and diabetes mellitus were factors associated with an increased risk of adverse outcome (Table 3).

### 3.3. SNPs in the CRP Gene as Prognostic Markers for Adverse CV Outcome

Tagging SNPs in the CRP gene (rs1130864, rs1417938, rs1800947, and rs3093077) were analyzed for their prognostic significance in relation to CV outcomes. In contingency analyses, no significant association of genetic variants to the occurrence of the combined endpoint was shown (Table 4). However, a worse CV outcome could be associated for carriers of the CT + TT genotype of rs1130864 (*p* = 0.089).

In further analyses of the prognostic potential of SNPs in the CRP gene on CV outcome, survival time was considered. Kaplan–Meier survival curves and log-rank tests were calculated for all SNPs. Indeed, one SNP, namely, rs1130864, clearly showed that the CT + TT genotype was associated with a less favorable CV prognosis (*p* _log rank_ = 0.041, HR: 1.26) (Figure 3).

However, when taking other factors influencing the prediction of CVD (age, male gender, BMI, smoking, diabetes mellitus, hypertension, and hyperlipoproteinemia) into account, the CT + TT genotype was not found to be an independent prognostic factor for adverse CV outcome (Cox regression is displayed in Table 5). In this analysis, age, male sex, and diabetes mellitus emerged as factors with the strongest significant predictive power for CV events.

## 4. Discussion

This longitudinal cohort study investigated the importance of selected genetic variants of the CRP gene (rs1130864, rs1417938, rs1800947, and rs3093077) and circulating CRP serum levels as predictors of CV events. For this purpose, a cohort of patients with angiographically proven CVD was selected and followed over a period of 10 years.

### 4.1. Circulating CRP Level as Prognostic Markers for Adverse CV Outcome

CRP level has been shown to be an independent risk factor of CVD as well as a significant predictor of CVD outcomes [8,35]. Therefore, CRP level has been regarded as a significant marker in CV risk stratification both in primary and secondary prevention [36].

As indicated in previous studies, the present study also confirmed that an elevated serum level of CRP is associated with an adverse CV outcome even after 10 years. A large-scale German study of patients with coronary artery disease (*n* = 13,100) supports this hypothesis in terms of 1-year survival [37]. With a longer follow-up period, too, this influence could also be confirmed in other studies (AtheroGene study: *n* = 1806, median follow-up time = 3.5 years, [38], Edinburgh Artery Study: *n* = 1592, follow-up period = 17 years [39]). As we already demonstrated in the 3-year follow-up, the prognostic significance of elevated CRP level independent of classical CV risk factors, such as age, male sex, elevated BMI, smoking, the presence of diabetes mellitus, hypertension, and hyperlipoproteinemia, remained in the 10-year follow-up [3-year follow-up: HR: 1.77; 95% confidence interval: 1.16–2.72; 10-year follow-up: HR:1.49; 95% confidence interval: 1.16–1.91) [21]. Considering additional CV prognostic factors, CRP could also be considered as an independent prognostic indicator for the progression of CVD in the studies by Ndrepepa [37] and in the Edinburgh Artery Study [39]. In contrast, elevated CRP was not identified as an additional prognostic outcome marker in addition to the classic CV risk factors in the AtheroGene study [38]. However, it must be noted that the AtheroGene study only considered a follow-up period of 1 year. Furthermore, there were still discrepancies regarding the endpoint examined (AtheroGene: CV death and MI; present study: cardiovascular death, death from stroke, MI, and stroke/TIA).

In other recent investigations, CRP was not considered the sole marker but rather in interaction with other factors. In a Chinese study, the importance of CRP to albumin ratio as an independent predictor of major adverse cardiac and cerebrovascular events and MI CVD patients was highlighted (*n* = 9375, follow-up period = 2 years) [40]. In another large-scale prospective cohort study, the elevation of both Lp(a) and CRP was associated with a significantly higher risk of major adverse cardiac and cerebrovascular events (*n* = 10,424; follow-up period = 5 years) [41]. The results of the various studies suggest that, in addition to the classic risk factors, other clinical parameters or combinations thereof might be used to stratify CV risk.

### 4.2. SNPs in the CRP Gene as Prognostic Markers for Adverse CV Outcome

A large number of studies have demonstrated that the level of CRP is also associated, among other factors, with genetic specificities in the CRP gene [20,42,43,44]. According to a recent study the number of genomic loci associated with circulating CRP levels was expanded to 266 [20]. Moreover, the percentage of variance explained by genetic predisposition improved to 16.3% [20]. The correlation between SNPs in the CRP gene and circulating CRP level was also shown in a clinical study we performed [21]. In this study, rs1130864, rs1800947, as well as rs1417938 were associated with CRP expression in CVD patients.

Factors that potentially influence CRP expression, which have been shown to be associated with circulating CRP levels, might also play an important role in the context of CVD. In a large-scale study with more than 500,000 participants, the importance of the genetic characteristics of CRP in terms of expression and involvement in the pathogenesis of a variety of inflammatory diseases was demonstrated [44].

Since CVDs are considered inflammatory diseases, it is reasonable to assume that SNPs in the CRP gene may also be associated with the incidence and/or prognosis of CV events. In a previous study, the CT + TT genotype of rs1130864 was proven to be an independent predictor for adverse CV events in a 10-year follow-up (Kaplan–Meier survival curve, log-rank test, *p* = 0.041). We were also able to show previously that CT + TT genotype of rs1130864 was associated with a significant increase in CRP expression in CV patients (CRP level: CC genotype 7.1 mg/L vs. CT + TT genotype 10.3 mg/L, *p* = 0.023) [21]. It is possible that the increased CRP level associated with CT + TT genotype is influential in adverse CV outcome at 10 years. The present data confirm the results of Sukhinina et al., who revealed the impact of CT + TT genotype of rs1130864 on unfavorable CV outcome in Russian CV patients during a 2-year follow-up [30]. Nevertheless, according to the potential impact of SNPs in the CRP gene to CV prognosis, the available data are inconsistent. Lange et al. proved in a clinical cohort of CV patients of white European descent that T-carriers of rs1417938 were at higher risk for stroke and CV mortality in a 3-year follow-up period [29]. Other SNPs studied (rs1800947, rs1205 and rs2808630) were not of prognostic significance. Furthermore, according to an Italian study, carriers of the GG genotype of rs1800947 were associated with an adverse CV prognosis at 2-year follow-up [31]. In contrast, other clinical studies in American men [32] and in Spaniards [33] did not establish a correlation between genetic features of the CRP gene and CV outcome.

Considering the controversial results of these studies, it is not recommended at the current time to include the genetic features of the CRP gene in the CV risk profile. The prospective studies conducted to date differ with respect to many factors, such as inclusion criteria, predefined CV endpoint, follow-up time, and ethnicity. It is therefore recommended to investigate the prognostic significance of SNPs in the CRP gene in large-scale prospective cohort studies considering different ethnicities.

### 4.3. Further Prognostic Markers for Adverse CV Outcome

In addition to primary prevention of CVD, secondary and tertiary prevention also plays an important role in public health [45]. A variety of classic risk factors affecting CV outcome have been described in this context [45]. To better assess the significance of both elevated CRP level and presence of CT + TT genotype of rs1130864 as prognostic factors, multivariate Cox regression analysis added classic risk factors for adverse CV outcome (Table 3 and Table 4). In the present study, we confirmed that CRP level and also genetic variant rs1130864 can indeed be considered as independent CV prognosis factors. Furthermore, according to the current literature and in the assessment of the American Heart Association and American College of Cardiology, it could be confirmed in the present study using a multivariate risk model that increased age, male sex, and the presence of diabetes are also of prognostic relevance (Table 3 and Table 4) [46]. Based on these results, we recommend considering a complex model when assessing new, possibly predictive factors for CVD.

### 4.4. Limitation of This Study

The present study was designed as a longitudinal cohort study. It was conducted to establish assumptions of the possible prognostic value of genetic variants of CRP and CRP level on adverse CV outcome during a 10-year follow-up period. The results of this study are representative for CV patients of Central Germany and cannot be generally extrapolated for the general population or other patient cohorts.

## 5. Conclusions

The present study indicates that an elevated level of CRP (≥5 mg/L) and presence of the CT + TT genotype of the rs1130864 variant in the CRP gene can be considered as predictors of adverse CV outcome in CV patients at 10-year follow-up. This association was confirmed for an elevated CRP level independent of the conventional CV risk factors. Considering the available data, CV patients with elevated CRP levels and the presence of a CT + TT genotype are at increased risk for future CV events and should therefore be monitored at closer recall intervals.

## Figures and Tables

**Figure 1 cells-12-01775-f001:**
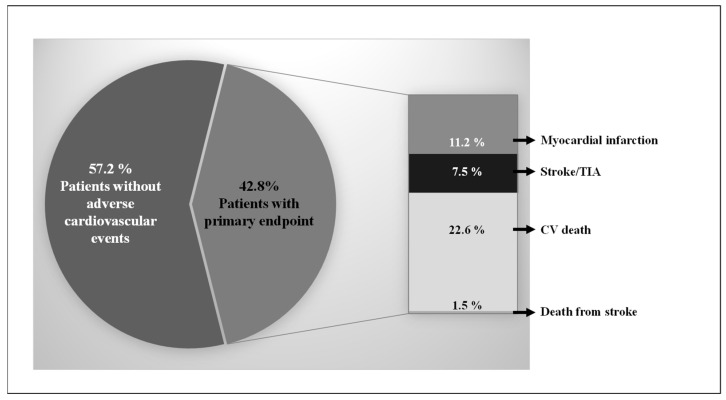
Distribution of the combined endpoint (cardiovascular (CV) death, death from stroke, myocardial infarction, and stroke/transient ischemic attack (TIA)).

**Figure 2 cells-12-01775-f002:**
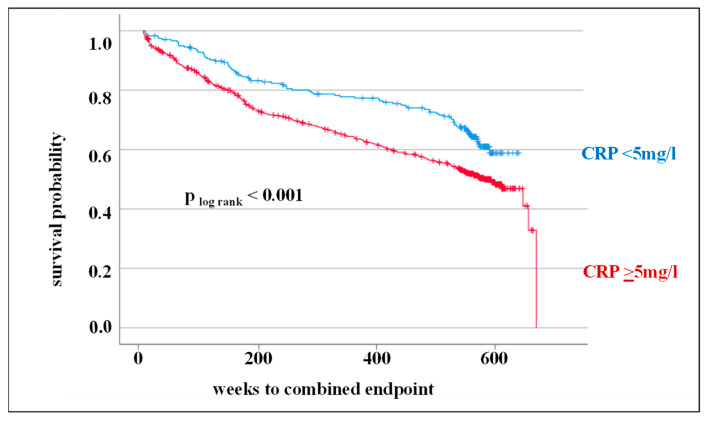
Kaplan–Meier survival curve and the log-rank test for the combined endpoint (cardiovascular death, death from stroke, myocardial infarction, and stroke/transient ischemic attack) considering circulating C-reactive protein (CRP) levels (<5 mg/L vs. ≥5 mg/L).

**Figure 3 cells-12-01775-f003:**
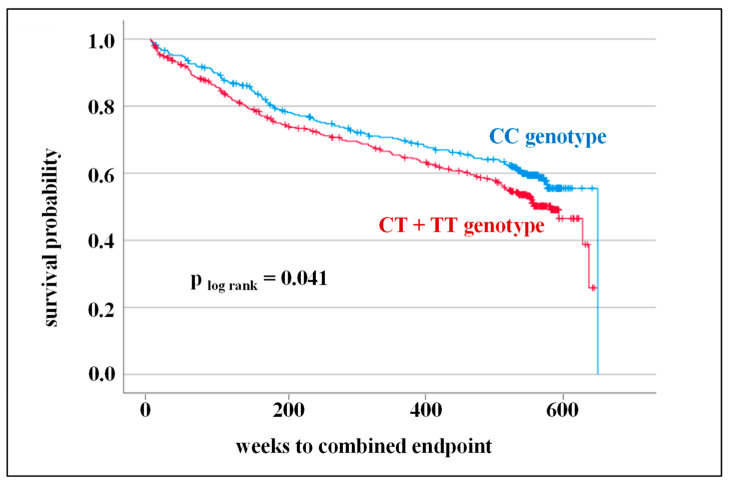
Kaplan–Meier survival curve and the log-rank test for combined endpoint (cardiovascular death, death from stroke, myocardial infarction, and stroke/transient ischemic attack) considering the genotype distribution of rs1130864.

**Table 1 cells-12-01775-t001:** Primers and restriction enzymes applied in polymerase chain reaction-restriction fragment length polymorphism (NG_013007.1).

SNP	Primer 3′–5′	Restriction Enzyme
rs1130864	Forward: cac gtc tct gtc tct ggt acct cc cgc	Mae II
	Reverse: caa aac acct ca aat tct gat tct ttt gga ac	
rs1417938	Forward: acc ccc at acct cag atc gaa	Tfi I
	Reverse: gac gtg acc atg gag aag ct	
rs1800947	Forward: cag ttt tac agt ggg tgg gtc	BsiHKAI
	Reverse: ccc gcc agt tca gga cat tag	
rs3093077	Forward: caa aag tga ggc tgg gac ctg	Mnl I
	Reverse: gac agg gag ctg aa gaga agg	

**Table 2 cells-12-01775-t002:** Baseline characteristics of cardiovascular patients with and without combined endpoint (cardiovascular death, death from stroke, myocardial infarction, stroke/TIA in 10-year follow-up). Skewed variables were evaluated by the Mann–Whitney U test and presented as the median (IQR: 25th/75th-interquartiles). Categorical variables are presented as a percentage and compared by chi-squared test. (IQR: interquartile range; TIA: transient ischemic attack; MI: myocardial infarction; HDL: high-density lipoprotein; LDL: low-density lipoprotein).

Characteristics	All Patients (*n* = 792)	Without Combined Endpoint (*n* = 453)	With Combined Endpoint (*n* = 339)	*p*-Value
Demographic and anamnestic parameters				
Age, years (median; 25/75 IQR)	69.2 (60.7/74.8)	68.2 (59.3/72.9)	70.9 (62.7/77.4)	<0.001
Female gender (%)	26.5	28.7	23.6	0.127 *
Current smoking (%)	10.4	11.5	8.8	0.278 *
Body mass index, kg/m^2^ (median; 25/75 IQR)	28.1 (25.3/30.7)	28.1 (25.3/30.9)	28.2 (25.4/30.7)	0.691
History of				
Diabetes mellitus (%)	35.6	29.4	44.0	<0.001 *
Hypertension (%)	87.6	86.1	89.7	0.160 *
MI (%)	39.3	32.5	48.4	<0.001 *
Stroke/TIA (%)	12.2	9.7	15.6	0.016 *
Peripheral artery disease (%)	10.2	7.3	14.2	0.002 *
Hyperlipoproteinemia (%)	59.0	59.2	58.7	0.955 *
Biochemical parameters (median; 25/75 IQR)				
C-reactive protein (mg/L),	8.6 (3.6/32.8)	6.2 (2.8/27.9)	11.4 (5.0/39.0)	0.001
Leukocytes (Gpt/L),	7.8 (6.4/9.6)	7.8 (6.4/9.8)	7.7 (6.2/9.5)	0.397
Interleukin 6 (pg/mL),	7.8 (3.8/15.9)	7.2 (3.4/13.9)	8.7 (4.3/18.5)	0.017
Uric acid (µmol/L)	5.7 (4.3/7.9)	5.4 (4.3/6.8)	6.3 (4.5/9.4)	<0.001
Creatinine (mmol/L),	87 (73/109)	83 (71/97)	97 (78/130)	<0.001
Total cholesterol (mmol/L),	4.3 (3.7/6.3)	4.4 (3.8/5.3)	4.2 (3.5/5.2)	0.030
HDL cholesterol (mmol/L),	1.0 (0.8/1.2)	1.0 (0.8/1.3)	0.9 (0.8/1.2)	0.002
LDL cholesterol (mmol/L),	2.5 (2.0/3.3)	2.6 (2.1/3.3)	2.5 (2.0/3.2)	0.055
Triglycerides (mmol/L),	1.4 (1.0/1.9)	1.4 (0.9/1.8)	1.3 (1.0/1.9)	0.394

* Yates correction.

**Table 3 cells-12-01775-t003:** Cox regression for assessment of the prognostic power of an increased C-reactive protein (CRP) level on the incidence of combined endpoint (cardiovascular death, death from stroke, myocardial infarction, and stroke/transient ischemic attack) considering classical cardiovascular risk factors as covariates. (CI: confidence interval; BMI: body mass index).

	Regression Coefficient	Standard Error	*p*-Value	Hazard Ratio	95% CI	
					Upper	Lower
CRP level ≥5 mg/L	0.397	0.129	0.002	1.49	1.16	1.91
Age	0.041	0.007	<0.001	1.04	1.03	1.06
Male gender	0.325	0.130	0.012	1.38	1.07	1.78
BMI	−0.008	0.014	0.534	0.99	0.97	1.02
smoking	0.317	0.209	0.128	1.37	0.91	2.07
Diabetes mellitus	0.540	0.118	<0.001	1.72	1.36	2.16
Hypertension	0.135	0.186	0.466	1.14	0.80	1.65
Hyperlipoproteinemia	−0.043	0.116	0.711	0.96	0.76	1.20

**Table 4 cells-12-01775-t004:** Association of single-nucleotide polymorphisms (SNPs) in the C-reactive protein (CRP) gene and combined endpoint without taking survival time into account.

SNPs in CRP Gene	All Patients (*n* = 792)	Without Combined Endpoint (*n* = 453)	With Combined Endpoint (*n* = 339)	*p*-Value
rs1130864				
CC genotype	44.4	47.2	40.6	
CT + TT genotype	55.6	52.8	59.4	0.089 *
rs1417938				
TT genotype	43.4	45.8	40.3	
AA + AT genotype	56.6	54.2	59.7	0.162 *
rs1800947				
GG genotype	86.5	85.4	87.9	
CG + CC genotype	13.5	14.6	12.1	0.370 *
rs3093077				
TT genotype	87.7	88.3	87.0	
GG + GT genotype	12.3	11.7	13.0	0.681 *

* Yates correction.

**Table 5 cells-12-01775-t005:** Cox regression for assessment of the prognostic power of single-nucleotide polymorphism rs1130864 on the incidence of combined endpoint (cardiovascular death, death from stroke, myocardial infarction, and stroke/transient ischemic attack) considering classical cardiovascular risk factors as covariates. (CI: confidence interval; BMI: body mass index).

	Regression Coefficient	Standard Error	*p*-Value	Hazard Ratio	95% CI	
					Upper	Lower
CT + TT genotype	0.188	0.116	0.103	1.21	0.96	1.51
Age	0.042	0.007	<0.001	1.04	1.03	1.06
Male gender	0.320	0.134	0.017	1.38	1.06	1.79
BMI	−0.007	0.014	0.062	0.99	0.97	1.02
Smoking	0.344	0.218	0.115	1.41	0.92	2.16
Diabetes mellitus	0.546	0.122	<0.001	1.73	1.36	2.19
Hypertension	0.102	0.192	0.597	1.11	0.76	1.61
Hyperlipoproteinemia	−0.114	0.119	0.339	0.89	0.71	1.13

## Data Availability

The comprehensive study protocol is available from the authors. Data can be provided by the authors upon request.

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
