# Peer review of "C-Reactive Protein Level and the Genetic Variant rs1130864 in the CRP Gene as Prognostic Factors for 10-Year Cardiovascular Outcome"

_cells, 2023, doi:10.3390/cells12131775_

Round 1
Reviewer 1 Report
Here are my comments on the manuscript by Schulz et al. entitled “C-reactive protein and its genetic variants as prognostic factors for 10 year cardiovascular outcome”
-The title should be restructured considering the variant rs1130864.
-In keyword section, please change genetics to genetic variants.
-In introduction section, the authors should define CRP, MI, and TIA. With respect to SNPs rs1130864, rs1417938, rs1800947, and rs3093077, please add a little more information about these SNPs, are they located in the promoter region of the CPR gene?
- Small nuclear polymorhpisms (SNPs) must be changed to single nucleotide polymorphisms (SNPs).
-In the next phrase (page 2, line 79-80), please change CV disease to CVD: “……cohort of patients with CV disease [21].”
-In materials and methods section, the authors should indicate the date of the blood extraction for CRP levels evaluation. Why was high-sensitivity C-reactive protein not considered to assess CRP levels?
- Biochemical parameters determination should be described in more details in materials and methods section.
-Please define the acronyms used in table 2 in the legend of table 2.
-The values of n should be homogenized: “n=9375, follow-up period=2 years” “n=10.424; follow-up period=5 years”
-In the discussion section, the authors should consider the effects of older age, male gender, and diabetes mellitus on the results.
Author Response
Dear Reviewer,
Thank you very much for your invaluable suggestions for revising our manuscript. We introduced your comments and critical remarks in the revised manuscript.
Here, we want to explain how the manuscript was revised. The changes are indicated using blue color.
The title should be restructured considering the variant rs1130864
The title of the manuscript has been revised as follows:
C-reactive protein level and the genetic variant rs1130864 in CRP gene as prognostic factors for 10-year cardiovascular outcome
In keyword section, please change genetics to genetic variants
Keyword section has been updated as proposed.
In introduction section, the authors should define CRP, MI, and TIA.
In the introduction, the abbreviations mentioned have been explained:
C-reactive protein (CRP), a marker of acute phase response, represents a very crucial parameter of inflammation involved in noncommunicable diseases, including CVD. … The evaluation of the study included the potential impact of SNPs of CRP gene (rs1130864, rs1417938, rs1800947, rs3093077) as well as CRP plasma level on a combined CV endpoint (CV death, death from stroke, myocardial infarction (MI), and stroke/ transient ischemic attack (TIA)).
With respect to SNPs rs1130864, rs1417938, rs1800947, and rs3093077, please add a little more information about these SNPs, are they located in the promoter region of the CPR gene?
In the introduction section information about the location of the SNPs was added:
Among these genetic markers are the tagging single nucleotide polymorphisms (SNPs) rs1130864 (3’-UTR, C/T), rs1417938 (intron, T/A), rs1800947 (L184L, G/C), and rs3093077 (noncoding, C/T).
Small nucleotid polymorhpisms (SNPs) must be changed to single nucleotide polymorphisms (SNPs).
Thank you very much for the advice. This has been revised accordingly.
In the next phrase (page 2, line 79-80), please change CV disease to CVD: “……cohort of patients with CV disease.”
This phrase has been revised as suggested.
In materials and methods section, the authors should indicate the date of the blood extraction for CRP levels evaluation.
On the day after inpatient admission and confirmation of inclusion and exclusion criteria, blood sampling was performed in the morning between 8am and 9am. Serum parameters (including C-reactive protein, applying the particle-enhanced immunological turbidity test), were assessed.
On the following day, the study-specific medical history was taken. Baseline variables (age, gender, smoking habits, and body mass index) and patient’s medical history (e.g., diabetes mellitus, hypertension, peripheral arterial disease, dyslipoproteinemia, and medication) were assessed.
Why was high-sensitivity C-reactive protein not considered to assess CRP levels?
In the study, CRP level was determined by applying a latex particle-enhanced immunological turbidity test. CRP levels >1mg/l are recorded in the central laboratory of the Medical Faculty of Martin Luther University. This test makes it possible to measure CRP levels in the range of HSCRP. CRP values of >5mg/l are assessed as pathological. In this study, a cut off of 5mg/l was used in the analysis of CRP level as a possible predictor of adverse CV outcome.
Biochemical parameters determination should be described in more details in materials and methods section.
We have revised the materials and methods section and described in more detail how we determined the biochemical parameters:
Biochemical parameters were assessed in the Central Laboratory of the Medical Faculty of the Martin Luther University Halle Wittenberg (C-reactive protein: latex particle-enhanced immunological turbidity test; leukocytes: flow cytometry; interleukin 6: electrochemiluminescence immunoassay; uric acid: enzymatic colorimetric assay using uricase; creatinine: kinetic color assay, Jaffé method; total cholesterol: enzymatic colorimetric assay; HDL cholesterol: homogeneous enzymatic colorimetric assay; LDL cholesterol: homogeneous enzymatic colorimetric assay; and triglycerides: enzymatic colorimetric assay).
Please define the acronyms used in table 2 in the legend of table 2.
Thank you very much for this advice. The acronyms have been defined in table 2.
The values of n should be homogenized: “n=9375, follow-up period=2 years” “n=10.424; follow-up period=5 years”
These statistics were explained in the discussion section. In this regard, two different studies were described that addressed the potential prognostic significance of CRP in interaction with other markers for future CV events. Study 1: n=9375, follow-up period=2 years; study 2: n=10,424; follow-up period=5 years. This section has been revised for better understanding.
In the discussion section, the authors should consider the effects of older age, male gender, and diabetes mellitus on the results.
Thank you very much for the valuable advice. As known from many studies and as we have shown in tables 3 and 4, both the CRP level and genetic variant in the CRP gene are not exclusive predictors for CV outcome. In our study, increased age, male sex, and also the presence of diabetes were shown to be important factors for increased CV risk (tables 3 and 4). The discussion addresses these considerations.
We hope we have satisfied the concerns of the reviewer with regard to the points marked.
With kindest regards
Susanne Schulz on behalf of all authors

Reviewer 2 Report
The clinical and scientific value of this paper is above average. It could be greatly improved by by professional editing that would shorten the paper and use more accurate language. I could not correct everything but here is the gist of my report and suggestions.
Line 37: represents à is.
Line 38: Why mention inflammasomes. Seems ectopic.
Line 41: Delete “It could be shown… “
Line 43: Rephrase “guidelines for guidance…”
Line 49: Delete Lipid lowering, every reader knows what a statin does. Mayb: Statins lower the plasma concentrations of atherogenic lipoproteins and CRP and improve CV prognosis.
Line 68: SNPs = Small --> Single
Figure 2: The curve for CRP < 5 mg/mLis faint, barely visible in the PDF.
Figure 3: Also, the curve for CC genotype is faint, barely visible in the PDF.
Conclusions: At a 10-year follow-up, elevated plasma CRP concentrations (> 5mg/l) and the CT+TT 314 genotype of the rs1130864 in the CRP gene predicted adverse CV outcomes in CV patients. These associations were independent of the traditional CV risk factors. Considering the available data, CV patients with elevated CRP levels and carrying the CT+TT genotype are at increased risk for future CV events and should be frequently monitored.
Needs editing for readability.
Author Response
Dear Reviewer,
Thank you very much for your invaluable suggestions for revising our manuscript. We introduced your comments and critical remarks in the revised manuscript.
We would like to thank the reviewer regarding his/her positive assessment of our manuscript. In the revision of our manuscript, we focused the text and shortened the introduction in particular.
The manuscript was proofread for English by native speaker Sherryl Sundell, who was managing editor of the International Journal of Cancer and who has many years of experience as a professional editor.
We revised all the words and phrases that were addressed by the reviewer. In addition, we revised Figures 2 and 3 and used different colors to distinguish the graphs.
We hope we have satisfied the concerns of the reviewer with regard to the points marked.
With kindest regards
Susanne Schulz on behalf of all authors

Round 2
Reviewer 2 Report
Much improved